# Susceptibility to Colorectal Cancer Based on HSD17B4 rs721673 and rs721675 Polymorphisms and Alcohol Intake among Taiwan Biobank Participants: A Retrospective Case Control Study Using the Nationwide Claims Data

**DOI:** 10.3390/jpm13040576

**Published:** 2023-03-24

**Authors:** Tzu-Chiao Lin, Min-Hua Chuang, Chia-Ni Hsiung, Pi-Kai Chang, Chien-An Sun, Tsan Yang, Yu-Ching Chou, Je-Ming Hu, Chih-Hsiung Hsu

**Affiliations:** 1School of Medicine, National Defense Medical Center, Taipei 114, Taiwanjeminghu@gmail.com (J.-M.H.); 2Division of Colorectal Surgery, Department of Surgery, Tri-Service General Hospital, National Defense Medical Center, Taipei 114, Taiwan; 3School of Public Health, National Defense Medical Center, Taipei 114, Taiwan; 4Institute of Molecular Medicine, National Tsing Hua University, Hsinchu 300, Taiwan; 5Data Science Statistical Cooperation Center, Institute of Statistical Science, Academia Sinica, Taipei 114, Taiwan; 6Department of Public Health, College of Medicine, Fu-Jen Catholic University, New Taipei City 242, Taiwan; 7Department of Health Business Administration, Meiho University, Pingtung County 912, Taiwan; 8Graduate Institute of Medical Sciences, National Defense Medical Center, Taipei 114, Taiwan; 9Health Service and Readiness Section, Armed Forces Taoyuan General Hospital, Taoyuan 325, Taiwan

**Keywords:** alcohol consumption, retrospective case control study, colorectal cancer, GWAS, biobank, medical claims dataset, CRC risk, CRC development, polymorphism

## Abstract

Colorectal cancer (CRC) is a major public health issue, and there are limited studies on the association between 17β-hydroxysteroid dehydrogenase type 4 (HSD17B4) polymorphism and CRC. We used two national databases from Taiwan to examine whether HSD17B4 rs721673, rs721675, and alcohol intake were independently and interactively correlated with CRC development. We linked the Taiwan Biobank (TWB) participants’ health and lifestyle information and genotypic data from 2012 to 2018 to the National Health Insurance Database (NHIRD) to confirm their medical records. We performed a genome-wide association study (GWAS) using data from 145 new incident CRC cases and matched 1316 healthy, non-CRC individuals. We calculated the odds ratios (OR) and 95% confidence intervals (CI) for CRC based on multiple logistic regression analyses. *HSD17B4* rs721673 and rs721675 on chromosome 5 were significantly and positively correlated with CRC (rs721673 A > G, aOR = 2.62, *p* = 2.90 × 10^−8^; rs721675 A > T, aOR = 2.61, *p* = 1.01 × 10^−6^). Within the high-risk genotypes, significantly higher ORs were observed among the alcohol intake group. Our results demonstrated that the rs721673 and rs721675 risk genotypes of HSD17B4 might increase the risk of CRC development in Taiwanese adults, especially those with alcohol consumption habits.

## 1. Introduction

Colorectal cancer (CRC) ranks third in incidence and second in mortality of cancers worldwide [1]. It is the most common cancer and third leading cause of cancer-related deaths in Taiwan [2,3]. Several studies have explored CRC etiology because it is a major public health issue. In addition to age, sex, and a family history of CRC, obesity, a sedentary lifestyle, high red meat consumption, excessive alcohol consumption, tobacco use, and inflammatory bowel disease have been identified as driving factors behind increases in CRC incidence [4].

However, the Nordic Twin Study estimated that 40% of the variation in CRC risk could be attributed to heritability [5]. To date, approximately 19 single-nucleotide polymorphisms (SNPs) have been identified in genome-wide association studies (GWAS) to be associated with susceptibility for the development of CRC [6]. These genetic variants, along with high-penetrance germline mutations in known CRC susceptibility genes, including APC, MUTYH [7], POLE, and POLD1 [8], are associated with changes in the DNA mismatch repair, LKB1, SMAD4, BMPR1A, AXIN2, and TGFβR2 pathways, which are implicated in intestinal hamartomatous polyps and predisposition to cases resembling Lynch syndrome [9,10,11,12]. However, hereditary syndromes resulting from rare variations with high penetration account for less than 6% of CRC cases [13,14]. Numerous GWAS have been conducted for common low-penetrance variants to investigate the residual risks that determine predisposition to developing CRC [15]. 

Nevertheless, results from previous CRC GWASs carried out in populations with European ancestry may not be applicable to East Asian populations. Investigating different populations improves the validity of general risk variants of CRC, given how varying populations can show significantly different association strengths and allele frequencies [16]. Furthermore, CRC GWAS conducted on East Asians can help determine genetic risk variants specific for this population.

Large-scale epidemiological studies have consistently indicated a significant relationship between obesity, smoking, alcohol drinking, and CRC [17,18,19]. Further studies showed a possible time- and dose-dependent relationship between cigarette smoking and the risk of CRC. Smoking significantly increases the risk of developing CRC through the microsatellite instability pathway, characterized by high levels of microsatellite instability and hypermethylation of promoter CpG island sites. This leads to the inactivation of several tumor-suppressor genes and other tumor-related genes, such as mutations in the BRAF gene [20]. A study by Shaukat et al. found that a higher body mass index (BMI) was a risk factor for long-term colorectal cancer mortality. Reducing BMI could control the risk of cancer mortality [21]. Alcohol consumption is highly prevalent worldwide and is associated with a high incidence of CRC. In developed countries, alcohol consumption is a significant contributor to the development of CRC [22]. Alcohol and its metabolites have direct and indirect effects that promote carcinogenesis, leading to cancer formation. These effects result from genetic, epigenetic, biochemical, and immunological abnormalities [23,24]. 

Over the past decade, case–control and cohort studies have evaluated potential gene–environment interactions in CRC risk [25,26,27]. Choi et al. analyzed the UK Biobank database and found that adopting healthy lifestyle factors lowered CRC risk in populations with high genetic susceptibility [28]. However, because of the method of data collection, few studies have discussed utilization of the Charlson comorbidity index (CCI), which was used to reveal the general health statuses of the participants in our CRC GWAS. CCI is a scoring system based on weighting summary measures for clinically important concomitant diseases, in order to quantify an individual’s disease burden.

We conducted a retrospective case control study to investigate susceptibility variants associated with CRC risk in the Taiwanese population. Participants’ genotypic, lifestyle, and biochemical data were obtained from the Taiwan Biobank (TWB) project. We linked TWB participants’ data to their health care records in the National Health Insurance Research Database (NHIRD) to identify CRC and other comorbidities. Furthermore, we analyzed the associations between genetic variants and BMI, smoking, alcohol consumption, betel nut-chewing habits, and CCI score. To the best of our knowledge, this is the first report that used the TWB and the NHIRD to perform a CRC GWAS in Taiwan.

## 2. Methods

### 2.1. Data Sources

The data used in this study were derived from the TWB (which included health, lifestyle, and genotypic data for 2012–2018) and the NHIRD (longitudinal health insurance data for 2012–2018). The Taiwanese government sponsors the TWB, a population-based dataset that collects health, lifestyle, and genetic data from Taiwan residents [29]. The TWB aimed to recruit 200,000 community-based healthy participants aged 30–70 years, who had not been diagnosed to have cancer prior to 2024. In 2021, the TWB had blood sample data and physical examination results from 111,903 voluntary participants. 

All the participants completed a structured questionnaire regarding personal information, health, and lifestyle data through interviews with 29 recruitment offices. More than 22 million people participated in the National Health Insurance (NHI) program. To investigate the incidence of CRC, we extracted CRC and other comorbidity diagnostic codes from the NHIRD, which included registration files and data from original claims for reimbursement [30]. Diseases were coded based on the International Classification of Diseases, Ninth Revision, Clinical Modification (ICD-9-CM) code from 2006 to 2015, and ICD-10-CM code from 2016 to 2017 [30]. In addition, the NHIRD was linked to the Taiwan Cancer Registry (TCR) in order to identify cancer patients and categorize cancer types using the International Classification of Diseases for Oncology, Third Edition (ICD-O-3).

### 2.2. Ethical Considerations

The study protocol was reviewed and approved by the institutional review board (IRB) of the Tri-Service General Hospital (TSGH IRB No. A202105019). The TSGH IRB waived the requirement for signed informed consent based on the de-identified nature of the data from the TWB and the NHIRD.

### 2.3. Study Population

We initially recruited 111,903 TWB participants. We identified incident CRC patients with a primary diagnosis of CRC (ICD-O-3 codes C180-C189, C199, and C209) using the NHIRD from 1 January 2012 to 31 December 2018. In total, 145 patients with CRC incidence were recruited. To exclude bias from possible confounding factors, we performed propensity score matching using logistic regression with a matching ratio of 1:10. The regression model included the following covariates: age, sex, BMI, smoking, alcohol consumption, and betel nut-chewing habits. After matching, we enrolled 1450 non-CRC participants from the TWB.

### 2.4. Covariate Assessment

All TWB participants were asked about their alcohol, tobacco, and betel nut use habits using a structured questionnaire. We used inpatient and outpatient files from the NHIRD to determine whether the selected TWB participants had comorbidities. Details of the included comorbidities are shown in Appendix A. Furthermore, we used these comorbidities to calculate the CCI for each TWB participant to assess their general health status. 

### 2.5. Genotyping Data and Imputation

TWB participants’ DNA was extracted from their blood samples using QIAamp DNA blood kits (Qiagen, Valencia, CA, USA) according to the manufacturer’s protocol. The quality of the isolated genomic DNA was assessed using agarose gel electrophoresis and quantified with spectrophotometry [31]. Details of the SNP genotyping and imputation performed by the TWB have been reported previously [32]. SNP genotyping was performed using custom Taiwan BioBank chips and the Axiom Genome-Wide Array Plate System (Affymetrix, Santa Clara, CA, USA) at the National Center of Genome Medicine, Academia Sinica, Taipei, Taiwan. Custom Taiwan BioBank chips, which included 653,291 SNPs, were used to collect the genetic profiles of Taiwanese individuals [33]. During the GWAS, we excluded SNPs with minor allele frequency (MAF) < 5%, missing rate > 5%, or those in violation of the Hardy–Weinberg equilibrium (*p* < 10^−4^). Individuals with outlying autosomal heterozygosity rates (beyond a range of mean ± 3 standard deviations) and highly correlated individuals with identity-by-descent >0.1875 were also excluded from the subsequent analyses. 332,733 SNPs and 1316 TWB participants without CRC were retained after conducting quality control procedures [34]. SNPs at locus 5q23.1 (chr5:119,500,00–119,700,00) and 11p11.2 (chr11:45,600,000–45,800,00) were fine-mapped using SHAPEIT and IMPUTE2 (V2.3.1) based on eastern Asian (EAS) population data from Phase 3 (V5) of the 1000 Genomes Project as a reference panel, whereas imputed SNPs with call rate < 0.99, MAF < 1%, or HWE *p*-value of <1.0 × 10^−4^ were eliminated. Finally, we selected candidate SNPs for eQTL analysis, which were calculated using the tissue-specific all SNP-gene associations dataset in sigmoid and transverse colon tissues from the Genotype-Tissue Expression (GTEx) Portal database V8 (https://www.gtexportal.org/, accessed on 8 May 2022) [35]. 

### 2.6. Statistical Analyses

Analyses were performed using SAS (version 9.4; SAS Institute, Inc., Cary, NC, USA) provided by the Academia Sinica Branch of the Ministry of Health and Welfare Data Welfare Center. We performed chi-square tests and t-tests to examine the differences between groups in discrete and continuous variables. Furthermore, the cumulative incidence of CRC was generated using the Kaplan–Meier method, and log-rank tests were performed to examine the difference between these two curves.

Logistic regression analysis, with adjusted additive models, was performed to determine the odds ratios (ORs) of individual SNPs associated with CRC. We adjusted for the effects of age, sex, BMI, and the first 10 principal components (PCs). The logistic regression model included 10 principal components as covariates to control for population stratification. To further examine the effects of covariates on the SNP–CRC association, we performed a logistic regression and adjusted for smoking, alcohol consumption, betel nut-chewing habits, and CCI score (genome-wide significance was implemented with *p* < 5 × 10^−8^). To assess the interaction between lifestyle and gene factors, we implemented subgroup analyses, stratified by covariates. All analyses were performed using PLINK version 1.90, R packages, and SAS. All *p*-values were two sided, and *p* < 0.05 was considered statistically significant.

We analyzed all the data in April 2022, and the preferred statements followed by the reporting items of the STROBE statement of the case–control study are available as Appendix A.

## 3. Results

In this study, the incidence rate of CRC in the TWB participants was observed to be 129.6 per 100,000. The mean follow-up time of the CRC and non-CRC groups was 1.87 and 2.85 years, respectively. Among the 1461 TWB participants, there were no differences in age, sex, BMI index, smoking, alcohol consumption, and betel nut-chewing habits between the individuals with newly diagnosed CRC (145) and non-CRC individuals (1316). In addition, the CCI score of the CRC group was significantly higher than that of the non-CRC group (Table 1).

We conducted a GWAS of CRC using 1461 samples obtained from the TWB. The genomic control inflation factor was 1.01. The SNP found to be most significantly associated with CRC was SNP rs721673 (A > G, aOR = 2.62, *p* = 2.90 × 10^−8^). SNP rs721673 achieved nominal genome-wide statistical significance at *p* < 5 × 10^−8^ in the intron region of the 17β-hydroxysteroid dehydrogenase type 4 (HSD17B4) gene on chromosome 5 at 5q23.1. We used the NHGRI-EBI GWAS Catalog [36] to confirm the novelty of this variant, and the results suggest that the present GWAS is the first to detect a genome-wide significance level CRC-related variant in the HSD17B4 gene. Imputation was performed to search for additional CRC-associated functional SNPs, based on linkage disequilibrium information. We imputed the SNP rs721673 within a ±1 Mb region using 1000 genomes from the EAS population as a reference. After imputation, we found that SNP rs721675 (chr5:119611700, human hg38 assembly) in chromosome 5q23.1 (A > T, aOR = 2.61) was the most significantly associated with CRC (*p* = 1.01 × 10^−6^). To determine the functional basis of the two most significant SNPs (rs721673 and rs721675) at 5q23.1/HSD17B4, eQTL analyses was retrieved from the GTEx database [35]. We found that rs721673 and rs721675 were cis-acting eQTL for HSD17B4 expression in colon tissue (*p* = 3.2 × 10^−4^), with a normalized effect size of 0.098. The G risk allele of rs721673 and T risk allele of rs721675 increased HSD17B4 expression.

Next, we used the Kaplan–Meier method to measure the cumulative incidence of CRC (Figure 1). These results demonstrated that both the rs721673 risk genotypes AG/GG and rs721675 risk genotypes AT/TT were correlated with significantly higher CRC rates than the two wild genotypes (log-rank test *p* < 0.001 and *p* < 0.001, respectively). We further examined the association of HSD17B4 rs721673 and rs721675 with CRC by age, sex, BMI, smoking, alcohol consumption, betel nut-chewing habits, and CCI scores. The ORs for rs721673 and rs721675 were significantly higher in the alcohol consuming group than in the non-alcohol consuming group (aOR = 15.2, 95% CI = 3.91 to 58.8; aOR = 16.4, 95% CI = 3.33 to 83.3, respectively) (Table 2). However, other covariates did not influence the association between HSD17B4 rs721673/rs721675 and CRC.

## 4. Discussion 

Evidence on the association between HSD17B4 rs721673 and rs721675 polymorphisms and CRC is limited. We linked the data of TWB participants aged 30–70 years to the NHIRD to conduct this retrospective case control study. This is the first study to evaluate the association between rs721673 and rs721675 genetic variants and CRC risk, to the best of our knowledge.

According to the GTEx database, rs721673 AG/GG and rs721675 AT/TT are risk genotypes that promote HSD17B4 expression. In this study, the two SNPs increased CRC development risk in the Taiwanese population, especially in the alcohol-consuming group. The results suggest that rs721673 and rs721675 polymorphisms and alcohol consumption may play a role in CRC risk among Taiwanese individuals. 

The HSD17B4 gene codes for the HSD17B4 protein, also known as D-specific bifunctional protein and multifunctional protein 2. HSD17B4 is a 80 kDa multifunctional enzyme localized in peroxisomes, that contains three distinct functional domains [37]. In normal cells, HSD17B4 plays essential roles in the catabolism of long-chain fatty acids [38], branched fatty acids, and steroid hormones [39]. In the last decade, HSD17B4 has been reported to be involved in the tumorigenesis and progression of many cancers by promoting estrone production, which is subsequently metabolized into carcinogenic 4-OH or 16α-OH estrone metabolites [40,41]. Increased levels of 4-OH estrone can cause cancer by reacting with DNA to form depurination adducts, which in turn causes damage to specific genes involved in carcinogenesis or induces microsatellite instability [42,43].

Our findings demonstrate that risk groups with rs721673 and rs721675 may have increased HSD17B4 expression, and consequently, higher risks of developing CRC. We observed that these groups had a higher CRC incidence rate. This finding is similar to a previous study that reported an association between the dysregulation of estrogen-related pathways and CRC development [44], and that estrogen-related gene polymorphisms were positively correlated with CRC risk [45,46]. Overexpression of HSD17B4 has been observed in various cancers, including prostate cancer [47], hepatocellular carcinoma [48], and breast cancer [49]. Furthermore, increased expression of HSD17B4 was correlated with poorer prognosis in patients with prostate, breast, and colon cancers [50].

Interestingly, this study revealed that alcohol consuming patients with the HSD17B4 rs721673 (AG/GG) and rs721675 (AT/TT) risk genotypes had a higher risk of CRC development than non-alcohol consuming patients. Many studies have shown that alcohol consumption is a significant risk factor for CRC development [19,51,52] and is estimated that alcohol increases the chance of CRC by by 52–60% [53,54]. 

Even in small quantities, alcohol consumption has been suggested to be linked to an increased risk of CRC. The connection between alcohol intake and CRC risk is dependent on the dose. Heavy drinking is mainly associated with a higher CRC risk, while the risk of CRC from light to moderate drinking remains inconsistent. A recent meta-analysis of the literature from 1966 to 2013 reveals an overall relative risk (RR) of 1.21 (95% CI = 1.01 to 1.46) for individuals who consume 56.5 g/day of alcohol. Furthermore, gender can influence the impact of alcohol on CRC risk; for those who drink 12.5–50 g/day, males exhibit a CRC RR of 1.10 (95% CI = 1.03 to 1.18), while females have a RR of 0.87 (95% CI = 0.65 to 1.16) [55]. Alcohol metabolites, including reactive oxygen and nitrogen species, can cause genetic, epigenetic, biochemical, and immunological dysfunction that promote chronic inflammation and cancer development [23,53]. Cytochrome P450 2E1 (CYP2E1) plays an important role in ethanol metabolism, which produces reactive oxygen species resulting in a vicious cycle of carcinogen accumulation [56]. Both animal and human studies have reported that chronic alcohol intake can increase CYP2E1 levels [57,58]. In addition, alcohol and its metabolites can influence gene expression in CRC by modifying the levels of particular miRNAs (microRNAs). By adjusting the relative quantities of specific miRNAs, ethanol can indirectly affect processes such as lipid metabolism, epithelial to mesenchymal transition (EMT), angiogenesis, and immune response, consequently altering carcinogenesis [59]. An example of miRNAs that are disrupted by ethanol is miR-34a, which is well-known as a tumor suppressor and is directly regulated by p53. MiR-34a is involved in numerous processes advantageous to the carcinogenic state in the colon. Under normal circumstances, miR-34a regulates hepatic glucose, lipid, and drug metabolism. In colorectal cancer cases, it is generally downregulated [60].

Although we did not find evidence indicating that smoking, BMI, and betel nut-chewing habits may modify the relationship between HSD17B4 rs721673 (AG/GG), rs721675 (AT/TT), and CRC. However, other literature has indicated these lifestyle factors may be associated with the development of CRC. A meta-analysis of 188 studies published between 1958 and 2018 found that current smokers are 1.14-fold more likely to develop CRC than non-smokers, and former smokers are 1.17-fold more likely [20]. Cigarette smoke contains carcinogenic substances, such as polycyclic aromatic hydrocarbons and N-nitrosamines, which can lead to the formation of DNA adducts and oxidative DNA damage. These genetic alterations can contribute to the initiation and progression of CRC [61]. BMI is a widely recognized indicator of obesity, and its relationship with CRC has been the subject of extensive research. The mechanisms linking obesity to CRC are complex and involve multiple pathways. A state of chronic low-grade inflammation characterizes obesity. Furthermore, obesity is associated with insulin resistance and altered levels of adipokines and other cytokines, which can promote colorectal carcinogenesis [62]. In a recent systematic review, Li et al. revealed that individuals with overweight and obesity (BMI ≥ 25 kg/m^2^) have a greater risk of early-onset CRC in comparison to those with normal weight (OR = 1.42, 95% CI = 1.19 to 1.68) [63]. Betel nut chewing, a prevalent habit among particular Asian, Pacific, and South Asian populations, has drawn increased attention in recent years due to its connection to multiple health risks, particularly oral cancer and other oral health problems. The primary alkaloids in betel nuts, such as arecoline, arecaidine, guvacoline, and guvacine, can induce a range of systemic effects, impacting the nervous, cardiovascular, gastrointestinal, and endocrine systems. Current evidence from Taiwan also indicates that this habit may play a role in forming colorectal polyps [64].

To the best of our knowledge, only a few studies have discussed the association between estrogen-related pathways and alcohol consumption and CRC risk. Konstandi et al. reported that estrogens increased hepatic CYP2E1 mRNA expression in ovariectomized mice [65]. In addition, Liu et al. demonstrated that alcohol intake was associated with increased circulating estrone levels in pre-menopausal adult women in a breast cancer study [66]. Furthermore, several cell line studies have indicated that estrogen receptor pathways may be modified by ethanol [67].

Based on our results, we believe that the HSD17B4 rs721673 (AG/GG) and rs721675 (AT/TT) risk groups had increased HSD17B4 expression and CYP2E1 levels, which in turn promoted CRC carcinogenesis in alcohol consumers. This hypothesis needs to be substantiated by further studies to elucidate the underlying roles of HSD17B4 and alcohol consumption in colorectal carcinogenesis.

It is crucial for health hcare professionals and public health policy makers to prioritize alcohol cessation interventions to reduce the incidence of CRC. Encouraging healthier lifestyle choices, such as adopting a balanced diet, increasing physical activity, and decreasing alcohol consumption, can help lower the risk of CRC. Moreover, targeted public health campaigns and educational programs can raise awareness of the detrimental effects of alcohol consumption on colorectal cancer risk, especially in the group with HSD17B4 rs721673 (AG/GG) and rs721675 (AT/TT) risk genotypes.

## 5. Conclusions

This study has some limitations. First, the TWB restricted the participants’ age to 30–70 years; hence, we could not examine the data of those over 70 years of age. Second, although well-trained interviewers conducted face-to-face interviews with the TWB participants, there is a possibility of response bias as participants’ information was collected using questionnaires. Finally, compared with other GWAS, the number of CRC cases in this study was relatively low and may not have been sufficient to explore many other significant SNPs involved in CRC development. However, we used NHIRD population data to confirm new CRC cases in the TWB participants to decrease the potential bias. 

Our results demonstrated that the HSD17B4 rs721673 and rs721675 risk genotypes might increase the risk of CRC development in Taiwanese adults, especially those with alcohol consumption habits. However, this finding needs to be substantiated with further studies to elucidate the role of HSD17B4 rs721673 and rs721675 and the modifying effect of alcohol consumption in colorectal carcinogenesis. 

## Figures and Tables

**Figure 1 jpm-13-00576-f001:**
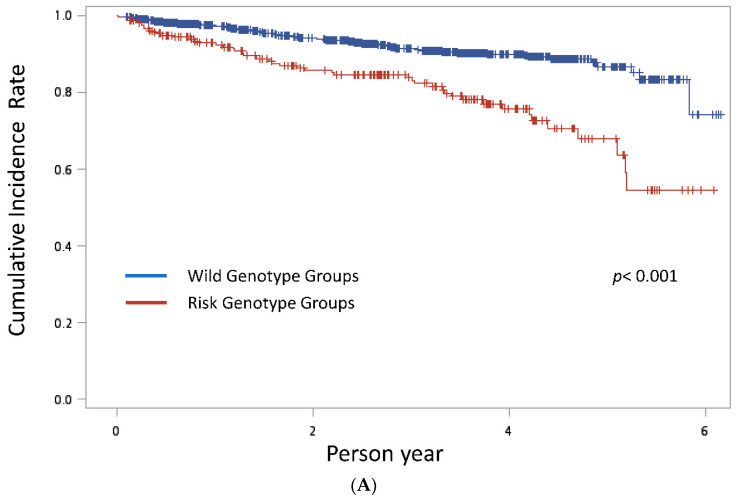
(**A**) Kaplan–Meier estimates of the cumulative incidence rate of CRC with HSD17B4 rs721673 risk genotype and wild genotype groups. (**B**) Kaplan–Meier estimates of the cumulative incidence rate of CRC with HSD17B4 rs721675 risk genotype and wild genotype groups.

**Table 1 jpm-13-00576-t001:** Demographic and clinical characteristics of study participants.

	CRC (*n* = 145)	Non-CRC (*n* = 1316)	*p*-Value
Age, years [mean ± SE]	58.41 ± 7.64	58.27 ± 7.7	0.835
Male sex [No. (%)]	66 (45.52)	606 (46.05)	0.973
BMI, kg/m^2^ [mean ± SE]	24.91 ± 3.10	25.02 ± 3.56	0.685
Smoking [No. (%)]	41 (28.28)	393 (29.86)	0.763
Alcohol [No. (%)]	15 (10.34)	139 (10.56)	1.000
betel nut-chewing [No. (%)]	18 (12.41)	182 (13.83)	0.731
CCI [mean ± SE]	1.37 ± 2.13	0.56 ± 0.96	<0.001 *

* *p* < 0.05. CCI, Charlson comorbidity index; CRC; colorectal cancer.

**Table 2 jpm-13-00576-t002:** Multivariate logistic regression analysis for the association between HSD17B4 rs721673 and rs721675 and the risk of colorectal cancer (CRC) based on sex, age, and BMI groups.

Covariates	rs721673 (A > G)		rs721675 (A > T)	
	* aOR(95%C.I.)		* aOR(95%C.I.)	
Age				
<50	2.72	(1.89–3.91)	1.73	(0.51–5.85)
≥50	1.73	(0.51–5.85)	2.75	(1.81–4.15)
Sex				
Male	2.64	(1.67–4.17)	2.27	(1.34–3.85)
Female	2.58	(1.54–4.31)	2.99	(1.69–5.26)
BMI				
BMI < 24	2.59	(1.48–4.52)	2.37	(1.29–4.37)
24 ≤ BMI < 27	2.89	(1.63–5.1)	3.24	(1.67–6.29)
BMI ≥ 27	2.58	(1.3–5.1)	2.61	(1.18–0.01)
Alcohol consumption				
NO	2.3	(1.6–3.31)	2.26	(1.5–3.39)
Yes	15.15	(3.91–58.82)	16.39	(3.33–83.33)
Smoking				
NO	2.45	(1.64–3.66)	2.42	(1.54–3.83)
Yes	3.3	(1.7–6.41)	3.32	(1.59–6.94)
Betel nut-chewing habits				
NO	2.95	(1.74–3.61)	2.51	(1.66–3.8)
Yes	3.53	(1.28–9.71)	2.95	(0.92–9.43)
CCI score				
CCI = 0	2.15	(1.26–3.67)	2.06	(1.16–3.67)
CCI ≥ 1	1.76	(1.82–4.67)	3.07	(1.78–5.30)

* In addition to the covariate that was used to confirm the effect of the association between SNPs and CRC, the other covariates, including the first 10 principal components (PCs), were adjusted for odds ratios.

## Data Availability

Data are available on request from the corresponding author due to restrictions, e.g., privacy or ethics. The data are not publicly available due to the Informed Consent Statement.

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
