# Peer review of "Susceptibility to Colorectal Cancer Based on HSD17B4 rs721673 and rs721675 Polymorphisms and Alcohol Intake among Taiwan Biobank Participants: A Retrospective Case Control Study Using the Nationwide Claims Data"

_jpm, 2023, doi:10.3390/jpm13040576_

Round 1

Reviewer 1 Report

This is a high-quality study on an important issue in colorectal carcinogenesis. The estrogen-related pathways belong to the relatively less known areas in colorectal cancer research. This paper identified the importance of two allelic polymorphism in a gene (hydroxysteroid 17 beta dehydrogenase 4) along the estrogen pathway, based on the genetic data from Taiwan Biobank.

The methods are appropriate and well-described, the study was carefully designed and performed. The authors took care on possible confounders and other issues which might arise in relation to such a molecular epidemiological study.

The manuscript is well written, results are appropriately presented, and the discussion is relevant.

Author Response

This is a high-quality study on an important issue in colorectal carcinogenesis. The estrogen-related pathways belong to the relatively less known areas in colorectal cancer research. This paper identified the importance of two allelic polymorphism in a gene (hydroxysteroid 17 beta dehydrogenase 4) along the estrogen pathway, based on the genetic data from Taiwan Biobank.

The methods are appropriate and well-described, the study was carefully designed and performed. The authors took care on possible confounders and other issues which might arise in relation to such a molecular epidemiological study.

The manuscript is well written, results are appropriately presented, and the discussion is relevant.

Thank you for your comment.

Reviewer 2 Report

I have several concerns aiming at giving more precisions of MS.

1. In the title of the MS, it could be clear that this is a case-control study.

2. Did the authors adhere to the STOBE Statement-Checklist for case-control studies? https://www.strobe-statement.org/ 

3. What were the inclusion criteria and exclusion criteria, respectively?

4. Statistical analysis, were all data normally distributed? if not, what statistical test were used?

5. In table 1, "Data are presented as means ± standard error (SE) or percentages and SE. However, some data". However, some data appear to be presented as the median (IQR). For example: smoking 41(28.28), alcohol 15(10.34) and betel nut chewing 18(12.41). This point should be clarified.

6. The quality of the figures could be improved.

7. What is the log-rank p in the graphics? This must be reported.

8. Figure legends need to be improved.

Author Response

  1. In the title of the MS, it could be clear that this is a case-control study.

Thank you for your suggestion. We have revised the Title of our manuscript, and please see the revised Title of Front Matter.

  1. Did the authors adhere to the STOBE Statement-Checklist for case-control studies? https://www.strobe-statement.org/

Thank you for your suggestion. The reporting items of the STROBE statement of the case-control study are available as Supplemental Table 2, and please see the revised manuscript in Methods on page 7.

  1. What were the inclusion criteria and exclusion criteria, respectively?

Thank you for your question. We initially recruited 111,903 TWB participants. We identified incident CRC patients with a primary diagnosis of CRC (ICD-O-3 codes C180-C189, C199, and C209) using the NHIRD database from January 1, 2012 to December 31, 2018. In total, 145 patients with CRC incidence were recruited. To exclude bias from possible confounding factors, we performed propensity score matching using logistic regression with a matching ratio of 1:10. The regression model included the following covariates: age, sex, BMI, smoking, alcohol consumption, and betel nut chewing habits. After matching, we enrolled 1450 non-CRC participants from the TWB. Please see the manuscript in Methods on page 5-6.

  1. Statistical analysis, were all data normally distributed? if not, what statistical test were used?

Thank you for your comment. According to the Central Limit Theorem, with a sufficiently large sample size, the distribution of the sample mean will be approximately normal. We recruited 1461 subjects (145 CRC patients and 1316 non-CRC participants) in this study, and the case number is fit the assumption of the Central Limit Theorem. So, we did not perform the Shapiro-Wilk test to check the normality assumption. Based on your suggestion, we performed the Shapiro-Wilk test and found the age, BMI, and CCI score data are non-uniform distributed. Therefore, we grouped these three variables according to Table 2 and conducted a chi-square test to examine if there is a significant difference in distribution between the case and control groups. The results showed that there was no statistically significant result between the two groups in terms of age and BMI, but there was a significant difference in CCI score. This result is consistent with the results of the t-test we conducted previously.

  1. In table 1, "Data are presented as means ± standard error (SE) or percentages and SE. However, some data". However, some data appear to be presented as the median (IQR). For example: smoking 41(28.28), alcohol 15(10.34) and betel nut chewing 18(12.41). This point should be clarified.

Thank you for your comment and we have revised Table 1. Please see the revised manuscript in Table 1.

  1. The quality of the figures could be improved.

Thank you for your comment and we have improved the quality of Figure 1. Please see the revised manuscript in Figure 1.

  1. What is the log-rank p in the graphics? This must be reported.

Thank you for your comment and we have added log-rank p value in Figure 1. Please see the revised manuscript in Figure 1.

  1. Figure legends need to be improved.

Thank you for your comment and we have revised the Figure 1 legends. Please see the revised manuscript in Figure 1.